# Dopaminergic Input Regulates the Sensitivity of Indirect Pathway Striatal Spiny Neurons to Brain-Derived Neurotrophic Factor

**DOI:** 10.3390/biology12101360

**Published:** 2023-10-23

**Authors:** Maurilyn Ayon-Olivas, Daniel Wolf, Thomas Andreska, Noelia Granado, Patrick Lüningschrör, Chi Wang Ip, Rosario Moratalla, Michael Sendtner

**Affiliations:** 1Institute of Clinical Neurobiology, University Hospital Wuerzburg, 97078 Wuerzburg, Germany; ayonolivas_m@ukw.de (M.A.-O.); wolf_d4@ukw.de (D.W.); thomas@andreska.de (T.A.); lueningsch_p@ukw.de (P.L.); 2Instituto Cajal, Consejo Superior de Investigaciones Científicas (CSIC), 28002 Madrid, Spain; ngranado@cajal.csic.es (N.G.); moratalla@cajal.csic.es (R.M.); 3CIBERNED, Instituto de Salud Carlos III, 28002 Madrid, Spain; 4Department of Neurology, University Hospital Wuerzburg, 97080 Wuerzburg, Germany; ip_c@ukw.de

**Keywords:** BDNF, TrkB, cortico-striatal synapse, basal ganglia, indirect pathway, DRD2, iSPNs, synaptic plasticity, GPCR, cholinergic interneurons, Pitx3

## Abstract

**Simple Summary:**

Motor dysfunction is among the main symptoms of Parkinson’s disease (PD). This is closely linked to the loss of the neurotransmitter dopamine (DA) in the midbrain and in the striatum. Additionally, the ability of striatal neurons to undergo adaptive cellular alterations and synaptic plasticity is impaired. Dopamine receptor D1 (DRD1) stimulation is needed for the establishment of long-term potentiation (LTP) at synapses of striatal spiny projection neurons (SPNs), which leads to enhanced neurotransmission. In contrast, dopamine receptor D2 (DRD2) stimulation is needed for the formation of long-term depression (LTD) in SPNs, which leads to the opposite effect. The tropomyosin receptor kinase B (TrkB) and its ligand brain-derived neurotrophic factor (BDNF) are centrally involved in plasticity regulation at the corticostriatal neurons synapses. There are two populations of striatal SPNs, with different projection targets in the brain. DRD1 is expressed in direct pathway spiny projection neurons (dSPNs) and its activation enhances TrkB sensitivity for BDNF by increasing the levels of TrkB at the cell surface. In this study, we showed that the activation of DRD2 in cultured striatal indirect pathway spiny projection neurons (iSPNs) and cholinergic interneurons causes the retraction of TrkB from the plasma membrane. This provides an explanation for the opposing synaptic plasticity changes observed upon DRD1 or DRD2 stimulation. In addition, TrkB was found within intracellular structures in dSPNs and iSPNs in *Pitx3*^−/−^ mice, a genetic model of PD with early onset dopaminergic depletion in the dorsolateral striatum (DLS). This dysregulated BDNF/TrkB signaling might contribute to the pathophysiology of direct and indirect pathway striatal projection neurons in PD.

**Abstract:**

Motor dysfunction in Parkinson’s disease (PD) is closely linked to the dopaminergic depletion of striatal neurons and altered synaptic plasticity at corticostriatal synapses. Dopamine receptor D1 (DRD1) stimulation is a crucial step in the formation of long-term potentiation (LTP), whereas dopamine receptor D2 (DRD2) stimulation is needed for the formation of long-term depression (LTD) in striatal spiny projection neurons (SPNs). Tropomyosin receptor kinase B (TrkB) and its ligand brain-derived neurotrophic factor (BDNF) are centrally involved in plasticity regulation at the corticostriatal synapses. DRD1 activation enhances TrkB’s sensitivity for BDNF in direct pathway spiny projection neurons (dSPNs). In this study, we showed that the activation of DRD2 in cultured striatal indirect pathway spiny projection neurons (iSPNs) and cholinergic interneurons causes the retraction of TrkB from the plasma membrane. This provides an explanation for the opposing synaptic plasticity changes observed upon DRD1 or DRD2 stimulation. In addition, TrkB was found within intracellular structures in dSPNs and iSPNs from *Pitx3*^−/−^ mice, a genetic model of PD with early onset dopaminergic depletion in the dorsolateral striatum (DLS). This dysregulated BDNF/TrkB signaling might contribute to the pathophysiology of direct and indirect pathway striatal projection neurons in PD.

## 1. Introduction

Parkinson’s disease (PD) is the second most common neurodegenerative disorder and is characterized by a progressive loss of dopaminergic neurons in the substantia nigra (SN) [1,2,3,4]. This loss leads to a variety of symptoms and can include both motor and non-motor symptoms. The most prominent features of PD include bradykinesia, rigidity, and rest tremors [5,6,7].

Defective dopaminergic signaling from the SN to the basal ganglia plays a central role for these altered motor functions. Dopamine (DA) released from the nigrostriatal terminals onto spiny projection neurons (SPNs) in the dorsolateral striatum (DLS) modulates circuit activity and plasticity, including long-term potentiation (LTP) and long-term depression (LTD) at the glutamatergic synapses between the corticostriatal afferences and striatal SPNs [8,9,10,11,12,13,14,15,16]. There are two distinct subpopulations of SPNs, giving rise to two different projection pathways. One population expresses dopamine D1 receptor (DRD1), which projects to the SN pars reticulata and to the internal segment of the globus pallidus and is generally thought to initiate and facilitate movement [17,18,19,20,21,22,23]. D1 receptors support the induction of plasticity in direct pathway SPNs (dSPNs) at corticostriatal synapses in the context of motor learning [14,15,24,25,26,27,28,29,30]. In contrast, dopamine D2 receptor (DRD2)-expressing SPNs project to the external segment of the globus pallidus. It has been assumed that this pathway negatively modifies circuits from the basal ganglia to the cortical regions, leading to the suppression of movements [17,18,20,22,23,31,32,33]. Defective DA signaling onto D1- and D2-expressing SPNs can, therefore, lead to both a reduction in voluntary movements and an increase in involuntary movements.

The DRD1 and DRD2 types of DA receptors in SPNs belong to the family of G protein coupled receptors (GPCRs) and have different functions, depending on their coupled Gα subunit [34,35]. In dSPNs, DRD1 is coupled to an activating Gα_s/olf_ subunit and, upon DA binding, activates adenylyl cyclase 5 (AC5), which facilitates cyclic AMP (cAMP) production [36,37,38,39]. In indirect pathway SPNs (iSPNs), DRD2 is coupled to an inhibitory Gα_i/o_ subunit, which inhibits AC5 and cAMP production upon DA binding [40,41,42,43]. This opposing effect of DA on these two SPN types is also reflected in their ability to undergo synaptic plasticity changes. In dSPNs, proper DA signaling through D1 receptors is needed for the induction of long-term potentiation (LTP) at corticostriatal synapses [10,24]. Conversely, in iSPNs, DRD2 activation is a prerequisite for the induction of long-term depression (LTD) [10,13,24,44]. Given the vital role of DA for corticostriatal plasticity, it is not surprising that abnormal plastic changes contribute to the motor symptoms of PD.

The tropomyosin receptor kinase B (TrkB), together with its ligand brain-derived neurotrophic factor (BDNF), is important for the facilitation of LTP in neurons. This has been extensively studied in the hippocampus and, to a smaller extent, also in the striatum [45,46,47,48,49,50,51]. TrkB is expressed both in dSPNs and iSPNs and establishes plasticity changes at the corticostriatal synapses [52,53]. However, the induction of such long-term changes depends on additional parameters. BDNF, the ligand for TrkB, is released from cortical afferents and, to a lesser extent, from the SN [26,54,55]. In order to be accessible for activation through BDNF and subsequent phosphorylation, TrkB needs to be present at the postsynaptic site in striatal SPNs. Previous work has shown that TrkB is rapidly recruited from intracellular stores to the plasma membrane when cAMP levels are rising [56,57,58,59,60]. Thus, the number of TrkB receptors present on the cell surface of dendritic spines in DRD1-expressing dSPNs is dependent on DA inputs and on the activation of the D1 receptor [61].

Indirect evidence for the role of BDNF/TrkB signaling in SPN plasticity comes from the finding that a reduction in cortical BDNF leads to motor alterations and deficits in motor learning [26]. The failure to adapt to a new motor challenge upon BDNF reduction might thus reflect impaired plasticity through the BDNF/TrkB signaling axis on SPNs. BDNF supply is also needed for the formation and maintenance of SPNs, as a loss in BDNF leads to dendritic spine degeneration and a massive degeneration of the striatum [62,63,64,65]. Similarly, studies on the pathophysiology of Huntington’s disease (HD) have shown that BDNF synthesis and transport in corticostriatal projection neurons is essential for the proper functioning of motor circuits [66,67]. In HD, dendritic spines on both dSPNs and iSPNs become atrophic, and mice expressing mutant huntingtin also show deficits in motor learning [68,69,70,71].

When DA is depleted from the DLS, both the DRD1 and DRD2 types of SPNs exhibit altered excitability, in particular hyperexcitability, and a concomitant degeneration of dendritic spines [29,30,72,73]. In a 6-OHDA lesion mouse model of PD, iSPNs were only able to express LTP when subjected to a spike-timing-dependent plasticity (STDP) protocol, which induces LTD under physiological conditions [10,44]. Conversely, dSPNs were only able to induce LTD when subjected to a LTP-inducing STDP protocol [10,44]. This indicates that DA loss in the striatum leads to a favored LTP induction in iSPNs, providing a possible explanation for their overexcitability. In contrast, dSPNs favor LTD induction, which renders them less excitable and could therefore explain the imbalances in motor control in PD.

Unlike DRD1, which in the striatum is only expressed in dSPNs, DRD2 is not exclusively expressed in iSPNs but also in cholinergic interneurons (ChIs) [74,75]. These interneurons are heavily interconnected with both SPN types, generally decreasing dSPN activity and increasing iSPN activity through the M4 and M1 muscarinic acetylcholine receptors, respectively [76,77,78].

In contrast to the classic lesion models of PD, where toxins are used to acutely disrupt dopaminergic innervation of the striatum, there is also the *Pitx3* gene knockout model of chronic dopamine deprivation. *Pitx3* gene knockout mice lack about 70% of dopaminergic fibers from the SN pars compacta from early development onwards, which leads to a depletion of ~90% of DA in the DLS [79,80,81,82,83]. PITX3 is an essential transcription factor for the development and maintenance of dopaminergic neurons in the SN [81,84]. As a consequence, only a few nigrostriatal neurons are generated, which results in a massive reduction in dopaminergic afferents to the DLS and an increasing dorso-ventral gradient of dopaminergic innervation [79,85]. This model also exhibits dendrite and dendritic spine degeneration in both SPN types in a DA concentration-dependent manner [85], similar to what has been observed in MPTP-treated monkeys [72].

In order to understand TrkB distribution in response to DA and DA depletion in DRD2-expressing iSPNs, we utilized an in vitro model of purified DRD2-expressing striatal neurons, as well as *Pitx3*^−/−^ mice as a genetic in vivo model of early onset chronic striatal DA depletion. We showed that TrkB cell surface translocation is dysregulated after DRD2 activation in iSPN cultures, and that this correlates with reduced phosphorylated TrkB in response to BDNF. Dysregulated TrkB cell surface expression was also found in dSPNs and iSPNs in the striatum of *Pitx3*^−/−^ mice, indicating that altered BDNF/TrkB signaling is not restricted to DRD1-expressing dSPNs, but also occurs in DRD2-expressing iSPNs and thus could contribute to the pathophysiology of specific disorders, such as PD, Huntington’s disease, and dystonia.

## 2. Materials and Methods

### 2.1. Animals and Housing

#### 2.1.1. Mice

For cell culture, brain tissue was dissected from P1-3 DRD2-eGFP male and female mice ([TG(DRD2-EGFP)S118Gsat]; RRID: MMRRC_000230-UNC). DRD2-eGFP mice were bred on a FVB/NJ genetic background.

All experiments were approved by a license for animal testing (approval numbers 55.2-2531.01 76/11 and 55.2-2532-2-728) and performed in accordance with supervision through the local veterinary authority (Veterinaeramt der Stadt Wuerzburg) and Committee on the Ethics of Animal Experiments, i.e., Regierung von Unterfranken, Wuerzburg, Germany.

For IHC analyses, BAC-transgenic DRD1-td-Tomato mice (Tg(Drd1a-tdTomato)6Calak; RRID: IMSR_JAX:016204) were crossed with wild-type or with *Pitx3*^−/−^ mice (RRID: IMSR_JAX:000942), and brain tissue was dissected from adult (P56-P84) male and female mice. DRD1-td-Tomato x *Pitx3*^−/−^ mice were bred on a C57Bl6/J genetic background.

DRD1-td-Tomato x wildtype/*Pitx3*^−/−^mice were provided by Dr. Rosario Moratalla, Instituto Cajal, Consejo Superior de Investigaciones Cientificas (CSIC), 28002 Madrid, Spain. The BAC transgenic mice were used to identify the striatal direct and indirect pathway neurons. All experimental procedures were approved by the Cajal Institute Bioethics Committee and by the CSIC Ethics Committee and fulfilled the requirements of Spanish (RD 53/2013) and European Union (63/2010/EU) legislation.

All mice were genotyped via PCR analysis. Mice were housed in groups of 4–6 per cage, with a constant room temperature (21–22 °C) and 12 h light/dark cycle (lights on at 7:00 h) and were given free access to food and water. 

#### 2.1.2. Rats

The studies with rats were performed as previously reported [61]. In brief, we used male Sprague-Dawley rats (RRID: RGD_10395233), which were housed at 22–24 °C, 50–60% humidity, and 12 h light/dark cycle. The animals were kept two per cage in polycarbonate cages (38 × 22 × 20 cm) with environment enrichment by providing paper as a nesting material. The rats were habituated to the laboratory environment and handled for a minimum of five days before conducting experiments.

#### 2.1.3. Unilateral 6-OHDA Lesion

For this procedure, 6-OHDA injections were performed as previously described [61]. Briefly, a 6-OHDA solution was prepared by dissolving in 0.9% sodium chloride + 0.1% ascorbic acid. The solution was then unilaterally injected into the medial forebrain bundle of the rats at a concentration of 6-OHDA at 3.6 mg/mL. The tip of the cannula was placed according to the brain Atlas by Paxinos and Watson, with the following coordinates: AP-4.4 mm, L-1.2 mm, V-7.8 mm, IB-2.4 mm (2.5 mL) and AP-4.0 mm, L-0.8 mm, V-8.0 mm, and IB +3.4 mm relative to bregma. Then, 3 µL of the 6-OHDA solution was injected at a rate of 1 µL per minute. For post-operative analgesia, the rats were directly injected with tramadol (12.5 mg/kg) after surgery and treated with tramadol in their drinking water for 2 further days. Successful 6-OHDA lesioning was confirmed via immunostaining against tyrosine hydroxylase (TH) in the striatum of the animals. TH staining can be found in our previous study [61], as we used the brain tissue of the same animals that were already used there.

### 2.2. Cell Culture

Striatal primary cultures were prepared from P1–3 newborn pups, as described previously [61]. Briefly, the striatum from both hemispheres was dissected and collected in 250 µL of ice-cold HBSS (Thermo Scientific, Schwerte, Germany, #14170–138) per animal. Trypsinization was performed with 50 µL of 1% trypsin (Worthington, Lakewood, NJ, USA, #TRL3) and 10 µL of 1% DNase for 30 min at 37 °C, and then 50 µL of a trypsin inhibitor was added to stop the reaction (Sigma-Aldrich, Taufkirchen, Germany, #T6522-100MG). Subsequently, the tissue was collected in a 15 mL tube filled with neurobasal medium (Thermo Scientific, Schwerte, Germany, #21103049) containing B27 supplement (Thermo Scientific, Schwerte, Germany, #17504-044), Glutamax (Thermo Scientific, Schwerte, Germany, #35050-038), and penicillin/streptomycin. Three trituration and centrifugation steps were performed. The cells were either enriched via cell sorting (see below) or were then resuspended in an appropriate volume of medium supplemented with 5% horse serum (Linaris, Dossenheim, Germany, #SHD3250YK) and directly plated on Poly-L-ornithine (Sigma-Aldrich; Taufkirchen, Germany, #P8638-500MG)/laminin-111 (Thermo Scientific, Schwerte, Germany, #23017-015)-coated cell culture dishes. After 24 h, the medium was changed to a serum-free medium. 

### 2.3. Flow Cytometry Cell Sorting

Enrichment of primary DRD2-eGFP neurons was accomplished via flow cytometry-based cell sorting using the MACSQuant Tyto Cell Sorter (Miltenyi Biotec, Bergisch Gladbach, Germany, #130-103-931). First, the sample was prepared through dissection and dissociation of the striatum, as described above. Prior to sorting, around 2–2.5 mL of cell suspension was loaded into the cartridge. For this procedure, the cartridge (Miltenyi Biotec, Bergisch Gladbach, Germany, #130-106-088) was prepared by wetting all the channels interconnecting the chambers with NB media supplemented with serum. Additionally, 170 µL of medium was added to the positive collection chamber to provide a suitable and nurturing environment to the sorted DRD2-SPNs. Before adding the cell suspension into the input chamber, the cells were filtered using a 20 µm pre-separation filter (Miltenyi Biotec, Bergisch Gladbach, Germany, #130-101-812) in order to remove cell aggregates and large particles that could clog the channels and hinder the sorting. Gating was set up to detect the GFP-positive neuronal population. The sorting procedure was performed at 25 °C, using a mixing speed of 100 rpm and the automatic pressure adjustment. Thereafter, the purified neurons were resuspended in an appropriate volume of media supplemented with horse serum and immediately plated on polyornithine/laminin-111-coated cell culture dishes. The cell culture medium was completely replaced with a serum-deprived cell culture medium after 24 h to prevent any serum-related effect on the dopamine receptors or TrkB stimulation. Cell cultures were maintained for 3–14 days in vitro (DIV) before preparation for ICC and Western blotting (including surface biotinylation). 

### 2.4. Stimulation of DRD2-eGFP Neurons

Enriched DRD2-eGFP neurons were pharmacologically treated with 500 nM sumanirole maleate (Bio-Techne, Wiesbaden-Nordenstadt, Germany, #227) and/or stimulated with 10 ng/mL BDNF (homemade). After stimulation, the cells were washed with 1 × PBS before further processing. 

### 2.5. Immunocytochemistry

Primary striatal neurons (DIV5-DIV10) were washed once with 1× PBS and fixed with 4% paraformaldehyde (Sigma-Aldrich, #158127) in 1 × PBS (37 °C, pH 7.4) for 15 min at RT. Subsequently, they were washed twice with 1 × PBS for 5 min. Permeabilization and blocking was performed with 0.1% Triton X-100 (Sigma-Aldrich; #X100-1L) and 10% donkey serum (PAN-Biotech, Aidenbach, Germany, #P30-0101) in 1 × PBS for 1 h. For immunostaining, the following antibodies were applied in a blocking solution overnight at 4 °C: chicken anti-GFP (1:1000; Abcam, Berlin, Germny, #ab13970), goat-anti-mTrkB (1:1000; R&D Systems, Minneapolis, MN, USA, #AF1494), rabbit anti-DARPP-32 (1:1000; Cell Signaling Technologies, Danvers, MA, USA, #19A3), mouse anti-ß-III-tubulin (1:1000; R&D systems, Minneapolis, MN, USA, #MO15013), goat anti-ChAT (1:200–500; Millipore, Burlington, MA, USA, #AB114), rat anti-Lamp-1 (1:20; Developmental Studies Hybridoma Bank, Iowa City, IA, # 1D4B-S; RRID: AB_528127), and sheep anti-mouse SORCS-2 (1:500; R&D systems, Minneapolis, MN, USA, #AF4237). After washing three times with 1XPBS for 10 min, primary antibodies were detected with the following secondary antibodies (incubation: 60–90 min at RT): donkey anti-chicken Alexa Fluor 488 (1:800; Jackson Immunoresearch, West Grove, PA, USA, #703-545-155), donkey anti-mouse Alexa Fluor 647 (1:800, Jackson Immunoresearch, West Grove, PA, USA, # 715-605-151), donkey anti-rabbit Cy3 (1:800; Jackson Immunoresearch, West Grove, PA, USA, #711-165-152), donkey anti-rabbit DyLight550 (1:200–400; Thermo Fisher Scientific, Schwerte, Germany, # SA5-10039), donkey anti-goat Alexa Fluor 647 (1:1500; Jackson Immunoresearch, West Grove, PA, USA, #705-605-003), donkey anti-rat Alexa 405 (1:800, Jackson Immunoresearch, West Grove, PA, USA, 712-475-15), and donkey anti-sheep Alexa Fluor 647 (1:500; Jackson Immunoresearch, West Grove, PA, USA, #2340703). Nuclei were stained with 0.4 mg/mL 4′,6-diamidino-2-phenylindole (DAPI). Coverslips were again washed three times for 10 min with 1× PBS and the cells were mounted on Superfrost_Plus glass slides (25 × 75 × 1.0 mm; Thermo Scientific, Schwerte, Germany, #J1800AMNZ) using MERCK-FluorSave reagent (Merck, Darmstadt, Germany, #345789-20ML). For cell surface staining, after stimulation, the media was discarded and cold media with the goat anti-mTrkB (1:1000; R&D Systems, Minneapolis, MN, USA, #AF1494) antibody was added to the wells. The cells were incubated for 30 min at 4 °C. Afterwards, the cells were washed twice with ice-cold PBS and blocked for 1 h at RT. Then, incubation with the following other primary antibodies was performed: chicken anti-GFP (1:1000; Abcam; #ab13970), rabbit anti-DARPP-32 (1:1000; Cell Signaling; #19A3), mouse anti-GFAP (1:200; Millipore, Burlington, MA, USA, #MAB3402), and rabbit anti-GFAP (1:1000; GeneTex, Irvine, CA, USA, GTX108711).

### 2.6. Preparation of Tissue for Immunostaining

The mice were anesthetized with pentobarbital (50 mg/kg, i.p.) and then transcardially perfused with 4% paraformaldehyde dissolved in PB (phosphate buffer, pH 7.4), dissected, and the brains were immersed overnight in the same fixative solution. The brains were washed and kept in PBS until they were used. The brains were then embedded in 6% agarose (Biozym, Hessisch Oldendorf, Germany, #840004) and 40 mm free-floating, coronal brain sections were obtained using a LEICA Vibratome VT1000S (RRID:SCR_016495). Sections were stored in “Cryoprotection Anti-Freeze Buffer” (comprising 1X PBS, glycerol (Honeywell, Charlotte, NC, USA, #15523-1L), and ethylene glycol (Roth, Karlsruhe, Germany, #9516.1) at −20 °C. 

### 2.7. Immunohistochemistry

Immunohistochemistry was performed using a free-floating staining protocol previously described [61]. In brief, vibratome brain sections were washed with 1X PBS, blocked and permeabilized with 0.3% Triton X-100 (Sigma-Aldrich, Taufkirchen, Germany, #X100-1L), 0.1% Tween 20 (Sigma-Aldrich, Taufkirchen, Germany, # P1379-1L), and 10% donkey serum (PAN-Biotech, Aidenbach, Germany, #P30-0101) in 1X PBS for 2–3 h. Then, primary antibodies were added to the blocking/permeabilization solution for 3 d at 4 °C. The following primary antibodies were used: goat anti-TrkB (R&D systems, Minneapolis, MN, USA, #AF1494), rabbit anti-Met-Enkephalin (Immunostar, Hudson, WI, USA, #20065), and chicken anti-tyrosine hydroxylase (Abcam, Berlin, Germny, #76442). They were used at a concentration ranging from 0.5 µg/mL to 1.0 µg/mL. After primary antibody incubation, the brain slices were washed with wash buffer (0.1% Triton X-100 and 0.3% Tween 20 in 1 × PBS). Then, secondary antibodies were added for 2 h at room temperature to the blocking/permeabilization solution. The following antibodies were used: donkey anti-goat Alexa Fluor 647 (Jackson Immunoresearch, West Grove, PA, USA, #705-605-147), donkey anti-chicken DyeLight405 (Jackson Immunoresearch, West Grove, PA, USA, #703-475-155), and donkey anti-rabbit Alexa Fluor 488 (Jackson Immunoresearch, West Grove, PA, USA, #711-545-152). They were used at a concentration of 0.55–0.625 µg/mL. Afterwards, the sections were washed 3× with wash buffer and the nuclei were stained with 1 µg/mL DAPI in 1 × PBS for 5 min. Then, the sections were washed 2× with 1 × PBS, shortly rinsed with H_2_O, and mounted on SuperfrostPlus glass slides (Thermo Fisher Scientific, Schwerte, Germany, #J1800AMNZ) with FluorSave mounting reagent (Merck, Darmstadt, Germany, #345789-20ML).

### 2.8. The Cell Surface Biotinylation Assay

Cell surface proteins were biotinylated using the Pierce Cell Surface Protein Isolation kit (Thermo Scientific, Schwerte, Germany, #89881), according to the manufacturer’s instructions. Between 150,000 and 300,000 cultured neurons were required per condition. The neurons were plated on 12 mm glass coverslips and biotinylated on either DIV7 or DIV8. 

The cells were washed with 1× cold PBS and incubated with EZLINK Sulfo-NHS-SS-biotin at 4 °C for 30 min, followed by the addition of a quenching solution. The cells were then lyzed with lysis buffer containing a protease inhibitor cocktail (Roche, Basel, Switzerland, #04693159001), sonicated one time, and centrifuged. The cell lysate was applied to the NeutrAvidin agarose gel. The flow through (intracellular protein fraction) was saved for Western blotting. The biotinylated cell surface proteins in the NeutrAvidin agarose gel were pulled down with sodium dodecyl sulfate (SDS) sample buffer (pH 6.8) with 50 mM dithiothreitol (DTT). Each sample was mixed with Laemmli buffer, heated to 95–99 °C for 5 min, and centrifuged shortly before loading onto 5–11% polyacrylamide gel. 

### 2.9. SDS-PAGE and Western Blotting

Polyacrylamide gel electrophoresis was performed in an electrophoresis cell chamber (Bio-Rad, Hercules, CA, USA #1658004) at 70 V for the stacking gel; then, the voltage was changed to 120 V once the samples reached the separating gel. The proteins were then transferred via wet blotting onto polyvinyl difluoride (PVDF) membranes (Bio-Rad, Hercules, CA, USA, #1620177) in a blotting cell (Bio-Rad, Hercules, CA, USA, #1703930) for 1 h at 4 °C with 135 V and 35 mA in blotting buffer (25 mM Tris, 192 mM glycine, 1% SDS, and 20% methanol). The membranes were blocked for 1–2 h with 5% milk powder in 1× TBS-T at RT and then incubated with either primary goat polyclonal antibodies (including goat polyclonal anti-mTrkB (1:5000; R&D Systems, Minneapolis, MN, USA, #AF1494), rabbit polyclonal anti-phosphoTrkB (Tyr816) (1:5000; Millipore, Burlington, MA, USA, #3587088), and rabbit anti-phospho TrkB (1:5000; Cell Signaling Technologies, Danvers, MA, USA, #46215)), chicken polyclonal anti-Calreticulin (1:2000; Thermofisher; #PA1902A), or goat-anti DARPP-32 (1:2000; R&D systems, Minneapolis, MN, USA, #AF6259) overnight at 4 °C on a rocker. Following this step, incubation with horseradish peroxidase (HRP)-conjugated secondary antibodies was performed (with either anti-goat, anti-mouse, anti-chicken, or anti-rabbit antibodies (1:10,000)). 

### 2.10. Western Blot Analyses

For the relative quantification of the effects of sumanirole treatment on the surface distribution of TrkB, the signal densities in the intracellular fractions, the membrane protein fractions, and the intracellular protein fractions were normalized to those of the marker proteins for their respective fractions.

### 2.11. Image Acquisition

Image acquisition was performed with an Olympus FluoView 1000 confocal laser microscope (Olympus, Shinjuku, Tokio, Japan). The different color channels were visualized using lasers with the wavelengths of 405 nm, 488 nm, 559 nm, and 633 nm. The microscope was equipped with the following objectives: HC PL APO CS2 40× (oil NA: 1.30) and HC PL APO CS2 60× (oil, NA: 1.35). Acquisition was performed using the Olympus FV10-ASW 3.0 (RRID: SCR_014215) software (Olympus, Shinjuku, Tokio, Japan); z-stacks of different focal planes were obtained; images were saved in the Olympus.oib format and analyzed with ImageJ image processing software version 1.53t (NIH, Bethesda, Maryland, USA) (RRID: SCR_003070). For visualization, the z-stacks were projected as a maximum intensity projection. The line scan colocalization analysis was performed using ImageJ. Brightness and contrast modifications were performed according to the rules of good scientific practice and the changes were documented for every image that was processed.

#### Quantification of TrkB Clusters

For the quantification of TrkB clusters, the number of TrkB-positive puncta were manually counted by the experimenter. TrkB cluster numbers were then normalized to the investigated area. Allocation to the D1- or D2-SPNs was conducted via colocalization of the TrkB clusters with either td-Tomato or Met-Enkephalin, respectively. 

### 2.12. Quantification and Statistical Analysis

For all data used for quantification purposes, a normal distribution was assumed based on experience with similar kinds of experiments. For two groups, a two-sample *t*-test was performed, and for more than two groups a one-way ANOVA with Tukey’ multiple comparison test was applied. For data where a control group has been set to 100%, a one sample *t*-test was performed. For all tests, a confidence level of *p* ≤ 0.05 was set. For the statistical analysis, GraphPad Prism software version 6.01 (Graph Pad, Boston, MA, USA) (RRID: SCR_002798) was used. The type of each statistical test used has been indicated in the respective figure legends.

## 3. Results

### 3.1. Enrichment of DRD2 Spiny Projection Neurons and Identification of DRD2-Expressing Cholinergic Interneurons

To investigate DRD2-mediated effects on the translocation of TrkB in indirect pathway iSPNs, we developed and tested a protocol for the culture of flow cytometry-enriched DRD2 SPNs (Appendix A). Our sorting protocol allowed us to increase the yield of isolated iSPNs from 1% to 2% in primary culture to ~80%. We were also able to confirm the identity of DRD2-expressing SPNs through the expression of DARPP-32, a marker for spiny projection neurons (Figure 1A). The significant enrichment of iSPNs makes this cell culture system an optimal way for in vitro studies intended to investigate the different cellular characteristics and molecular pathways of SPNs expressing DRD2, facilitating biochemical and immunocytochemical analyses. 

Due to their important functional role in the control and modulation of SPNs in the striatum and the presence of DRD2 on their cell surface [86], we were able to identify the population of cholinergic interneurons in our enriched iSPN cultures. These neurons overlapped with the 7.9% of cells after sorting that were DARPP-32-negative but expressed GFP at the same time (Figure 1B). To further characterize these cells and confirm their specific identity based on their neurotransmitter expression, we stained these cells against choline acetyltransferase (ChAT). As a negative control for ChAT staining, we used dorsal root ganglionic sensory neurons, and spinal motor neurons as a positive control (Figure 1C). Mixed striatal cultures show ChAT immunoreactivity in some GFP-positive cells. Cholinergic interneurons can also be identified with the aid of morphological criteria, such as their bigger somata and more complex dendritic extensions (Figure 1D,E).

### 3.2. DRD2 Activation in iSPNs Decreases TrkB Cell Surface Translocation 

In a previous study, we found that DRD1 activation in dSPNs increases TrkB cell surface exposure by stimulating the transfer of this receptor from the endoplasmic reticulum to the cell surface [61]. To investigate whether the activation of D2 receptors in iSPNs also modulates cell surface TrkB expression and responsiveness to BDNF, we used enriched DRD2 cultures of iSPNs from DRD2-EGFP reporter mice. These neurons were cultured for a minimum of 7 days in order to reach maturation and were then treated with sumanirole maleate (500 nM), a DRD2 agonist, for different time courses. We hypothesized that the activation of Gα_i_-coupled receptors would induce the inhibition of adenylyl cyclase activity and a progressive downregulation in cAMP levels. This decrease in cAMP could prevent TrkB from being translocated to the cell surface. In order to study such effects, we performed surface biotinylation and cell surface immunostainings of TrkB in the presence or absence of sumanirole. Biotinylation of cell surface proteins of DRD2 neurons treated under different time courses showed that TrkB surface levels decreased after 10 min of treatment with sumanirole and decreased further over time for up to 24 h (Figure 2A,B and Appendix A) (a one sample *t*-test was performed. Control: mean 100% (n = 7); 10 min. Sum: mean 77.6%, *p* = 0.05 (n = 7);; 60 min. Sum: mean 75.8%, *p* = 0.19 (n = 4); 6 h. Sum: mean 31.5%, *p* = 0.08 (n = 2); Sum 24 h: mean 28.0%, *p* = 0.06 (n = 2)). Confocal imaging also showed that DRD2’s activation with sumanirole led to a marked decrease in TrkB on the cell surface in comparison to their DA-depleted controls (Figure 2C). This reduction was still seen after 5 d of continuous sumanirole treatment and was reversed when sumanirole was washed out for 2 h at this time point (Figure 2C, right panel). Reduced TrkB could result in lower sensitivity to BDNF compared to DA-depleted controls. As a next step, we investigated how the reduced cell surface levels of TrkB correlate with 10hosphor-TrkB levels after BDNF exposure. The induction of TrkB phosphorylation was performed by adding BDNF for 5 min to enriched DRD2 cultures that had previously been preincubated with sumanirole for different time courses. This resulted in a significant decrease in TrkB phosphorylation in samples acutely treated with sumanirole for 10 min compared to their controls (Figure 2D and Appendix A). Interestingly, at 60 min and 24 h of treatment with the DRD2 agonist, no further reduction in pTrkB was observed in comparison to the controls, but rather the phosphorylation of TrkB returned back to control levels. Quantification of TrkB phosphorylation levels showed a significant reduction after 10 min, but no further decrease and a reestablishment of basal phosphorylation levels after longer treatment periods (Figure 2E), despite reduced TrkB cell surface levels (a one sample *t*-test was performed. Control: mean 100% (n = 7); Sum + BDNF 10 min: mean 46.64, *p* = 0.006 (n = 7); Sum + BDNF 60 min: mean 95.37%, *p* = 0.807 (n = 3); Sum + BDNF 24 h: mean 114.2%, *p* = 0.728 (n = 3)). We also assessed the ratio of pTrkB to cell surface TrkB, as seen with the surface biotinylation experiments at 10 min, 60 min, and 24 h treatment of sumanirole. A gradual increase in pTrkB/surface TrkB was observed (Figure 2F).

### 3.3. DRD2 Activation Leads to the Removal of TrkB from the Cell Surface and Subsequent Lysosomal Degradation within the Cell Body 

TrkB is usually present on the dendritic spines of iSPNs that receive inputs from BDNF-positive corticostriatal afferences. We therefore investigated how the activation of DRD2 with sumanirole affects dendritic TrkB in cultures of enriched DRD2-expressing striatal neurons. Activation of D2 receptors by sumanirole rapidly reduced cell surface TrkB expression in the soma and dendrites of iSPNs (Figure 3A). Blocking autophagy with bafilomycin led to an accumulation of SORCS-2 and Lamp-1-positive structures, which was even more pronounced after prolonged treatment (4 h). This was apparent in the soma but not in the dendrites of iSPNs. At a smaller scale, treatment with sumanirole led to a prominent reduction in TrkB in dendritic processes within 10 min (Figure 3B). This was also observed for the sorting receptor SORCS-2, which interacts with TrkB and, thus, guides it to the cell surface. The presence of Lamp-1-positive late endosomal/lysosomal vesicles was not affected by the 10 min treatment with sumanirole. When we treated the cells with bafilomycin, which prevents the fusion of endosomes or autophagosomal structures with lysosomes, TrkB’s immunoreactivity and SORCS-2’s immunoreactivity were unchanged at 10 min of treatment in comparison to the controls, and no accumulation was observed, indicating that only a small fraction of TrkB is directly degraded by lysosomes in dendritic processes. After 4 h of bafilomycin treatment, dendritic TrkB and SORCS-2 immunoreactivities were reduced. Since bafilomycin blocks the fusion of endosomes with lysosomes, this reduction cannot be due to enhanced lysosomal degradation within dendrites, but rather reflects the removal of TrkB from dendrites via retrograde transport. The colocalization of TrkB with SORCS-2 was acutely reduced after 10 min of sumanirole treatment (Figure 3C), indicating that vesicles that bring TrkB to the cell surface are reduced by sumanirole treatment (mean CTR: 55.2%, 10 min. Sum: 39.9%; one-way ANOVA with Tukey’s multiple comparisons test, CTR vs. 10 min. Sum: *p* = 0.0594). Bafilomycin treatment for 10 min (mean: 39.1%; one-way ANOVA with Tukey’s multiple comparisons test, CTR vs. 10 min. Baf: *p* = 0.0445) or 4 h (mean: 34.5%; one-way ANOVA with Tukey’s multiple comparisons test, CTR vs. 4 h Baf: *p* = 0.0190) did not enhance the colocalization of TrkB with SORCS2, indicating that a reduced fusion of endosomes with lysosomes has no positive effect on the recycling of TrkB to the cell surface.

When we investigated the effect of sumanirole treatment on TrkB distribution in the cell body of DRD2-expressing neurons in enriched cultures of striatal iSPNs, we found a rapid uptake of TrkB (Figure 4A) and the formation of intracellular accumulations of TrkB that appeared to be similar as those that we have identified and described previously in DRD1-expressing direct pathway striatal neurons [61]. Similarly, as in DRD1-expressing neurons, these cluster-like structures, in which TrkB was present, were also closely associated with Lamp-1-positive lysosomal vesicles (Figure 4A,B). When fusion between the endosomal vesicles and lysosomes was blocked via prolonged periods of bafilomycin treatment, the formation of TrkB-positive clusters increased, and an increase in the association of these TrkB clusters with lysosomal Lamp-1-positive vesicles was observed after 10 min (mean CTR: 32.4% (n = 5), 10 min. Baf: 60.2% (n = 3); one-way ANOVA with Tukey’s multiple comparisons test, CTR vs. 10 min. Baf: *p* = 0.0011) and 4 h (mean: 61.2% (n = 5); one-way ANOVA with Tukey’s multiple comparisons test, CTR vs. 4 h Baf: *p* < 0.0001) of bafilomycin treatment (Figure 4B,C). Furthermore, sumanirole treatment alone for 10 min also enhanced the number of TrkB-positive lysosomes (mean: 48.9% (n = 5); one-way ANOVA with Tukey’s multiple comparisons test, CTR vs. 10 min. Sum: *p* = 0.0186), but the simultaneous treatment with bafilomycin and sumanirole could not further increase the bafilomycin-induced TrkB–lysosome association (mean: 63.8% (n = 2); one-way ANOVA with Tukey’s multiple comparisons test, CTR vs. 4 h Baf, 10 min. Sum: *p* = 0.0015). This indicates that TrkB is degraded by lysosomes in the cell soma. Taken together, these data demonstrate that TrkB is rapidly retracted from the cell surface both in the dendrites and cell soma after the activation of DRD2 with sumanirole, and that dendritic TrkB is retrogradely transported to the soma, where these TrkB-positive vesicles then fuse with lysosomes. 

### 3.4. Dopamine Depletion Causes TrkB Cluster Formation in Pitx3^−/−^ Mice

We previously showed that the subcellular distribution of TrkB in SPNs is altered upon dopamine depletion in a rat model in which 6-OHDA is unilaterally injected into the medial forebrain bundle [61]. In this model, TrkB forms insoluble aggregates, which cluster at the perinuclear space exclusively in dSPNs. To examine whether this aggregation also occurs in the *Pitx3*^−/−^ mouse model, we investigated TrkB distribution in striatal tissue slices from these mice at an age range from 6 months to 11 months. We found that TrkB clusters arose in the striatum of *Pitx3*^−/−^ animals (Figure 5A) and that the *Pitx3*^−/−^ striatal sections showed a significantly (mean *Pitx3*^+/+^ (n = 5): 14.4; mean *Pitx3*^−/−^ (n = 6): 28.9; unpaired *t*-test: *p* = 0.0376) higher number of these clusters than sections from wild-type control animals (Figure 5B). We further wanted to know in which cells these clusters occur. For the identification of dSPNs, we analyzed td-Tomato expression in DRD1-positive cells; for the iSPNs, we performed immunohistochemical staining for Met-Enkephalin. We found that TrkB clusters do not only occur in dSPNs but also in iSPNs in the *Pitx3*^−/−^ model (Figure 5C,D). This is in contrast to our previous findings with the 6-OHDA rat model [61]. The TrkB clusters in the *Pitx3*^−/−^ mice developed under a condition where dopamine has been depleted for a prolonged timespan, as they already develop dopamine depletion during early developmental stages. To see whether the localization of TrkB clusters changes during prolonged periods of dopamine depletion, we analyzed striatal slices of 6-OHDA-treated rats at different time points after the lesion. However, the restriction of TrkB clusters to dSPNs did not change within 4 months after the lesion (Figure 5E,F), and the iSPNs did not show intracellular clusters even after a prolonged period of 4 months after 6-OHDA injection (one-way ANOVA with Tukey’s multiple comparisons test: 2 wk p.l. dSPN vs. 2 wk p.l. iSPN: *p* = 0.0059; 1 m p.l. dSPN vs. 1 m p.l. iSPN: *p* = 0.0160; 4 m p.l. dSPN vs. 4 m p.l. iSPN: *p* = 0.0233). Also, there was no significant change in the number of TrkB clusters after prolonged DA depletion (one-way ANOVA with Tukey’s multiple comparisons test: 2 wk p.l. dSPN vs. 1 m p.l. dSPN: *p* = 0.9971; 2 wk p.l. dSPN vs. 4 m p.l. dSPN: *p* = 0.9576; 1 m p.l. dSPN vs. 4 m p.l. dSPN: *p* = 0.9988).

### 3.5. Prolonged Dopamine Depletion Increases BDNF Expression in the Motor Cortex of 6-OHDA-Treated Rats

The motor cortex is the main input to the striatum, and it has been shown that corticostriatal projections deliver BDNF to the striatal SPNs [26]. To see whether dopamine depletion changes BDNF expression patterns, we performed immunostaining against BDNF in cortical slices from 6-OHDA-treated rats. The number of BDNF-positive cells increased from 2 weeks to 4 months after 6-OHDA injection both ipsilateral and contralateral to the lesion (Figure 6A). This is apparent in layer II/III (one-way ANOVA with Tukey’s multiple comparisons test: 2 wk p.l. intact vs. 1 m p.l. intact: *p* = 0.0003; 2 wk p.l. intact vs. 4 m p.l. intact: *p* = 0.0007; 2 wk p.l. lesioned vs. 1 m p.l. lesioned: *p* = 0.0574; 2 wk p.l. lesioned vs. 4 m p.l. lesioned: *p* = 0.0047; 2 wk p.l. intact vs. 2 wk p.l. lesioned: *p* = 0.9374; 1 m p.l. intact vs. 1 m p.l. lesioned: *p* = 0.1132; 4 m p.l. intact vs. 4 m p.l. lesioned: *p* = 0.9921) as well as in layer V (one-way ANOVA with Tukey’s multiple comparisons test: 2 wk p.l. intact vs. 1 m p.l. intact: *p* = 0.0587; 2 wk p.l. intact vs. 4 m p.l. intact: *p* = 0.0410; 2 wk p.l. lesioned vs. 1 m p.l. lesioned: *p* = 0.1249; 2 wk p.l. lesioned vs. 4 m p.l. lesioned: *p* = 0.0076; 2 wk p.l. intact vs. 2 wk p.l. lesioned: *p* = 0.8630; 1 m p.l. intact vs. 1 m p.l. lesioned: *p* = 0.6138; 4 m p.l. intact vs. 4 m p.l. lesioned: *p* = 0.9976) of the motor cortex, both of which are areas that send projections to the striatum. Interestingly, there was a trend towards a reduced number of BDNF-positive cells in the motor cortex of the lesioned hemisphere when compared to the intact hemisphere from 1 month after the lesion onwards (Figure 6B).

## 4. Discussion

In this study, we provide evidence that the activation of DRD2 in cultured striatal iSPNs and cholinergic interneurons causes a rapid retraction of TrkB from the cell surface. As a consequence of short-term DRD2 activation, TrkB’s sensitivity to BDNF was decreased, which is reflected by lower pTrkB levels. Prolonged DRD2 stimulation, however, restored pTrkB to baseline levels, while surface TrkB is even more decreased. In addition, perinuclear TrkB clusters were found both in dSPNs and in iSPNs in the *Pitx3*^−/−^ mouse model which displays dopaminergic depletion in the DLS from early developmental stages onwards. In contrast, acute dopamine depletion in the 6-OHDA rat model led to TrkB cluster formation in dSPNs but not in iSPNs, even months after the lesion. 

We have previously shown that BDNF sensitivity in DRD1-expressing dSPNs can be modulated via TrkB cell surface expression through the activation of D1 receptors. Under pathological conditions in 6-OHDA-treated rats and in postmortem brains of patients with PD, the decrease in dopaminergic innervation leads to the formation of TrkB clusters in the perinuclear area of dSPNs [61]. 

In order to understand the role of DA signaling for BDNF/TrkB in DRD2-positive neurons and how DA denervation leads to dendritic and spine atrophy or degeneration of striatal spiny projection neurons in PD, we have developed a cell culture technique for the enrichment of DRD2 striatal neurons. The isolation of DRD2-expressing iSPNs allowed us to study specific mechanisms that occurred selectively in this neuronal population. These neurons were cultured under dopamine-depleted conditions, which enabled us to study the effect of different pharmacological agents that modulate DRD2 activity, while minimizing the effect of other possible variables interfering with dopaminergic signaling. 

We found that TrkB cell surface localization was rapidly reduced after treatment with sumanirole maleate, a highly selective agonist of D2 receptors. Under healthy conditions, dopamine (DA) release from nigrostriatal terminals onto SPNs in the DLS modulates circuit activity and long-term plastic changes, including long-term potentiation (LTP) and long-term depression (LTD) at the glutamatergic synapses between corticostriatal afferences and striatal SPNs [8,9,10,11,12,13,14,15]. TrkB’s translocation to the cell surface is prevented via the stimulation of Gα_i_, reducing AC activity and, in turn, decreasing cAMP levels. This results in a decreased export of TrkB from intracellular pools to the cell surface, and consequently, to an overall decrease in sensitivity for BDNF in iSPNs (Appendix A). This process complements LTP effects induced via the activation of DRD1 in dSPNs. A balance between the activity of these two striatal output pathways, direct and indirect, maintains the correct functional organization of the basal ganglia. Likewise, a proper DA input controls the plasticity mechanisms that maintain, in synergy, these two neuronal types in order to sustain effective motor functions. 

Here, we also showed that TrkB’s localization in DRD2 neurons can be further modulated. Chronic activation of D2 receptors in vitro was achieved by stimulating enriched DRD2 neurons in culture over a period of 24 h or longer. This led to a further reduction in TrkB receptors at the cell surface of DRD2 striatal neurons. In parallel, intracellular levels of TrkB increased. This suggests that not only was TrkB’s translocation to the cell surface inhibited, but that the existent surface receptors might have been internalized, consequently reducing sensitivity to BDNF after acute stimulation of D2 receptors. 

Altered TrkB phosphorylation was already observed at 5 min after treatment with sumanirole. Interestingly, at 60 min and 24 h of treatment with this DRD2 agonist, no further reduction in pTrkB was observed. At that time point, pTrkB levels increased again when related to the levels of cell surface-exposed TrkB. This could reflect a cell autonomous mechanism that is due to a sensitization of the TrkB receptors left at the cell surface after undergoing a long period of stimulation. Long-term D2 receptor stimulation with sumanirole could lead to a compensatory or adaptive mechanism specific for DRD2 striatal neurons against chronic D2 agonist exposure. 

This effect of enhanced TrkB phosphorylation, despite low levels of TrkB at the cell surface after long-term stimulation with a DRD2 agonist, might as well be explained by an additional non-cell-autonomous effect due to the presence of DRD2-expressing astrocytes in the sorted D2 receptor-expressing cells (Appendix A). Astrocytes have been reported to express DRD2, and the activation of DRD2 in astrocytes downregulates cAMP, thus upregulating CNTF expression, which is then released close to the dopaminergic terminals [87,88,89,90]. CNTF could enhance TrkB phosphorylation, despite the lower concentration of TrkB receptors at the cell surface. As a first step, the upregulation of CNTF expression functions as a protective factor for the SPNs [89]. Therefore, under dopamine-depleted conditions, where SPNs are found atrophic, DRD2 activation in astroglia will compensate for the dendritic and spine loss via CNTF upregulation. However, the lack of a dopamine input to the astroglia will eventually lead to the downregulation of CNTF expression and spine degeneration. Dopaminergic denervation in adult mice reduced CNTF mRNA levels by 60%, whereas systemic treatment with the DRD2 agonist quinpirole increased CNTF expression [87,89]. Despite these results, there is some controversy regarding the survival effects of DRD2 activation. The most accepted explanation for this is that the time of stimulation with DRD2 agonists and antagonists can lead to opposite effects, meaning that long-term and short-term treatments have different consequences as part of an adaptation mechanism. This same reason might explain why we also observed this paradoxical effect in TrkB surface levels and pTrkB levels in our results [87,88].

We have also shown that 7.9% of the DRD2-expressing sorted striatal neurons are cholinergic interneurons. These interneurons regulate the synaptic plasticity and excitability of striatal SPNs and are thought to induce inhibition in these cells. Additionally, cholinergic interneurons are characterized by a multiphasic change in activity, showing a pause in firing which is triggered by aversive and salient stimuli. These pauses have been associated with changes in DA neuron activity. Furthermore, a blockage of D2 receptors in the cholinergic interneurons inhibited the generation of the pause, and the upregulation of D2 receptors prolonged the pause, which is associated with delayed inhibitory learning [86]. The chronic activation of DRD2 interneurons might negatively affect inhibitory learning due to an increase in the pause and subsequent reduction in cholinergic signaling, dysregulating SPN inhibitory control and contributing to the motor complications in PD. This sheds light on new mechanisms that might help to explain why striatal learning in PD patients is dysfunctional rather than simply absent [10,91]. How dopamine depletion in PD and treatment with dopamine agonists specifically affects plasticity in interneurons and consequently also SPNs, however, needs further clarification.

Overall, many changes take place in SPNs during the course of PD pathophysiology. Long-term dopamine depletion from early developmental stages in *Pitx3*^−/−^ mice, acute depletion of dopaminergic afferents in 6-OHDA-treated rodents, as well as chronic levodopa therapy in PD models that induce dyskinesia have shown to induce changes in the morphology and synaptic activity of SPNs from both direct and indirect pathways [29,44,73,92,93]. It has been shown that a high dose of levodopa that induces dyskinesia restores spine density in iSPNs after dopamine denervation [73,93,94]. On the other hand, dopamine denervation did not affect spine density in dSPNs, but treatment with levodopa at either a low or high concentration that causes levodopa-induced dyskinesia (LID) resulted in decreased spine density [94]. These studies have shown that the activation of DA receptors and chronic treatment with agonists in SPNs might involve spine reorganization at corticostriatal synapses in SPNs from both the indirect and direct pathways, and the observed gradual decrease in TrkB receptors after chronic sumanirole treatment could lead to LID, suggesting an important role for D2 receptors in PD and its treatment.

Furthermore, we studied the effect of DA depletion in DRD2 striatal neurons and the subsequent alterations of TrkB expression under pathological conditions of chronic DA depletion in *Pitx3*^−/−^ mice. *Pitx3*^−/−^ or aphakia mice represent a genetic model of PD, in which most of the dopaminergic neurons from the substantia nigra do not develop properly. Consequently, dopaminergic innervation of the DLS is massively reduced from early developmental stages onwards. When we examined TrkB expression in striatal brain slices of adult *Pitx3*^−/−^ mice, we found that they showed an increase in perinuclear TrkB cluster formation compared to their wild-type littermates. These clusters were apparent in both dSPNs and iSPNs in the DLS. Interestingly, the aggregation of TrkB was not restricted to the dSPNs, as opposed to what we have previously shown for 6-OHDA-treated mice and rats [61]. We systematically reexamined the slices of 6-OHDA-treated rats using Met-Enkephalin as a marker for iSPNs. This confirmed that upon acute DA depletion in the 6-OHDA model, TrkB clusters almost exclusively form in dSPNs, with no clusters in the Met-Enkephalin-positive iSPNs.

A possible explanation for this finding is the differences between the two models. In the 6-OHDA model, dopaminergic denervation occurs rapidly within a few days after the 6-OHDA injection, while in the *Pitx3*^−/−^ mouse model, the animals already lack striatal dopaminergic inputs during early development. This intrinsic lack of DA in the DLS could lead to developmental compensatory mechanisms counteracting potentially harmful changes in the motor network. This is also reflected by the absence of an apparent motor phenotype in these animals [73]. Thus, TrkB cluster formation happens in a state where the striatal network is adapted to missing DA and where there is already a long-term absence of dopamine receptor activation. In contrast, the DA depletion in the 6-OHDA model happens in a state where the striatal network is already fully developed, not allowing for huge compensatory changes. Thus, neurons in the DLS of 6-OHDA animals are more severely affected by DA depletion than those of the *Pitx3*^−/−^ mice.

It is apparent that TrkB clusters form upon DA depletion in both models, but iSPNs are spared from TrkB cluster accumulation in the 6-OHDA model. Potentially, dSPNs are more vulnerable to DA depletion than iSPNs, which could explain that TrkB cluster formation appears exclusively in the dSPNs in this model. This could be due to the distinct signal transduction processes of the different dopamine receptors. Under physiological conditions, D1 receptor activation leads to TrkB surface expression by shuttling TrkB from the intracellular stores to the cell surface. When this action is missing, TrkB can no longer be transported to the cell surface and is retained in the cell soma, favoring TrkB cluster formation. Conversely, D2 receptor activation normally leads to decreased TrkB surface expression. When this mechanism is impaired through DA depletion, TrkB surface expression is favored, which is likely facilitated via still intact A_2A_ receptor signaling. This would render iSPNs less prone to TrkB cluster formation, as intracellular TrkB levels are decreased compared to dSPNs in a DA-depleted state. In the *Pitx3*^−/−^ mice, there might be a compensatory downregulation of A_2A_ receptors to protect the iSPNs from excitotoxic signaling, thus reducing this protective effect on TrkB cluster formation. 

Furthermore, in the striatum, there is also a SPN population which expresses both DRD1 and DRD2 simultaneously [18,95,96]. However, we could not observe any TrkB clusters in the population which simultaneously expressed Met-Enkephalin and DRD1-tdTomato in *Pitx3*^−/−^ mice. This might be due to the low abundance of these DRD1/DRD2-coexpressing cells. Also, we cannot exclude the possibility that these cells might exhibit mixed responses to DA depletion, leading to a more complex effect on TrkB translocation. We did not investigate this in the 6-OHDA rat model, as D1-SPNs could not be directly identified by suitable markers, making it impossible to identify DRD1-expressing neurons that also express DRD2 or Met-Enkephalin.

Interestingly, BDNF expression in the cortex increased gradually after 6-OHDA injection both on the ipsilateral lesioned side and the contralateral non-lesioned side. As BDNF itself is crucial for motor learning and execution [26], and since the motor cortex is heavily connected to the striatum, it is possible that BDNF levels increase as a consequence of altered activity in response to striatal DA depletion. However, this response could also involve the contralateral non-lesioned side, since the lesioned hemisphere and the intact one are connected through interhemispheric projections that pass through the corpus callosum [97]. It is therefore possible that after DA depletion, enhanced activity takes place in neurons on the non-lesioned side that project to the contralateral lesioned hemisphere. Consequently, the motor cortex of the intact side would start to become involved in motor control of the lesioned side. Similar findings have been made in stroke patients with a unilateral lesion in their motor areas [98,99].

## 5. Limitations of the Study

There are several limitations in our study. We identified the presence of DRD2-expressing cholinergic interneurons in our cell culture model, but how the activation of DRD2 with sumanirole specifically in these cells contributes to TrkB internalization in this and other striatal cell populations has not been studied. For this, further investigations with striatal organoid or organotypic slice cultures are needed for being able to investigate the responses of these interneurons in a synaptic context with D1 and D2 dopamine receptor-expressing SPNs. Also, the 6-OHDA rat model and the Pitx3^−/−^ model are not directly comparable due to their different natures and degrees of DA depletion within the striatum. The conclusion from one model cannot be directly transferred to the other. However, we see these two models rather as complementary to each other to investigate how DA depletion affects striatal pathology either in an acute or in a chronic manner. Furthermore, the number of animals used for the quantification of the 6-OHDA experiments was low (n = 3). This was accounted for using non-parametric statistical testing when the data were assumed to be not normally distributed. Western blot analyses showed strong tendencies; however, due to low n numbers, no statistical significance could be reported in some cases.

## 6. Conclusions

In summary, we found that the activation of D2 receptors in iSPNs reduces TrkB cell surface levels. Upon retraction from the cell surface, TrkB accumulates in intracellular clusters and associates with Lamp-1-positive lysosomal structures. Biotinylation experiments indicated that this reduction in TrkB increases over time, suggesting a role for iSPNs in the complications induced by chronic treatment with levodopa. Interestingly, TrkB cell surface reduction did not affect TrkB phosphorylation levels after long-term treatment with a DRD2 agonist, which might explain why previous studies have reported that high doses of levodopa that induced dyskinesia rescue spine morphology in iSPNs. Additionally, using a *Pitx3*^−/−^ model of early onset dopaminergic depletion in the dorsolateral striatum, we found TrkB clusters within intracellular structures in the iSPNs and dSPNs. Thus, dendritic and spine atrophy correlate with altered TrkB cell surface expression, indicating that dysregulated BDNF/TrkB signaling contributes to the pathophysiology of direct and indirect pathway striatal projection neurons in PD, and that long-term stimulation of DRD2 might re-establish TrkB phosphorylation to basal levels, rescuing spine morphology. This regulatory mechanism might be crucial for positive or negative modulation of corticostriatal synaptic plasticity. Altogether, these findings provide new insights about the disease pathomechanisms and possible therapeutic targets.

## Figures and Tables

**Figure 1 biology-12-01360-f001:**
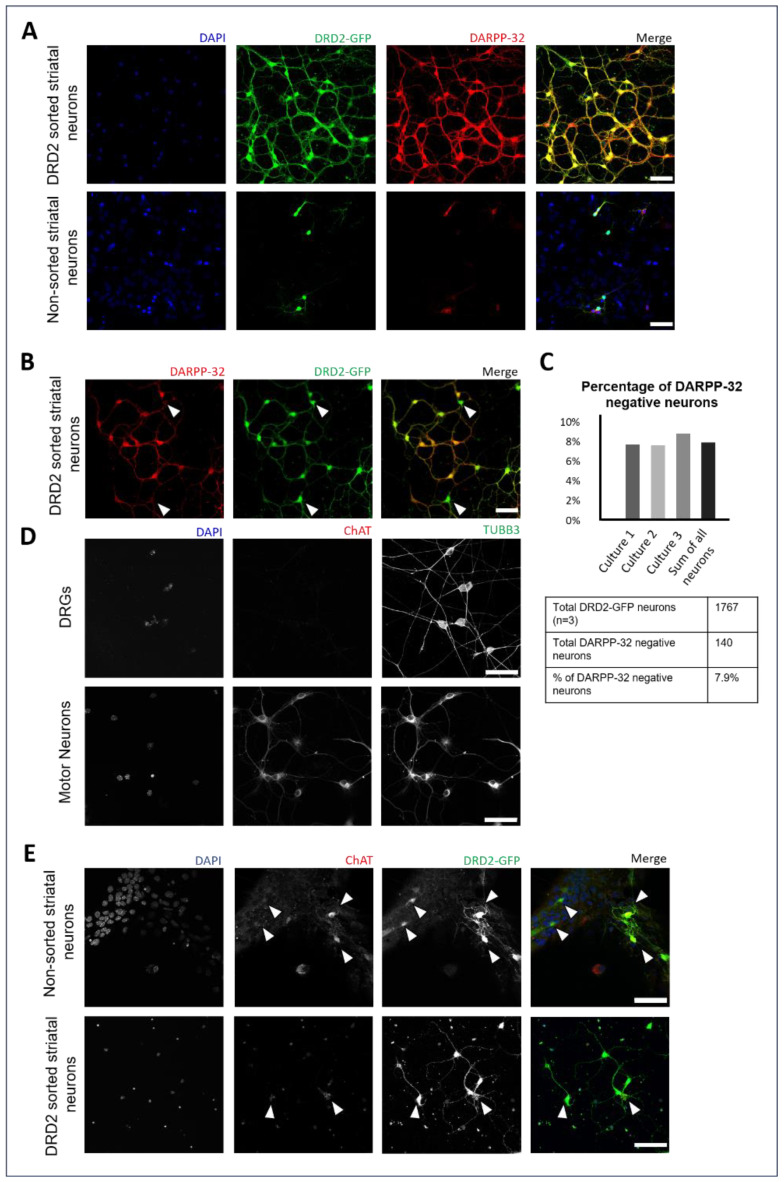
Enrichment of DRD2 neurons and identification of DRD2-expressing interneurons. Fluorescence-based sorting of primary striatal neurons expressing D2 dopamine receptors. (**A**) Immunocytochemistry of spiny projection neurons (DIV7) expressing DRD2-eGFP, DARPP-32, and DAPI (upper panel) compared to non-sorted primary striatal neurons (lower panel). (**B**) Identification of DRD2 interneurons which do not express DARPP-32. (**C**) Quantification of DRD2-eGFP-positive and DARPP-32-negative neurons indicates a 7.9% presence of DRD2 interneurons in sorted cultures (n = 3). (**D**) Dorsal root ganglionic sensory neurons (DRGs; upper panel) and spinal motor neurons (lower panel) as negative and positive controls for ChAT expression, respectively. (**E**) Non-sorted striatal cultures (upper panel) and sorted DRD2 neurons (lower panel) showing GFP-positive cholinergic interneurons. Scale bars: 50 µm.

**Figure 2 biology-12-01360-f002:**
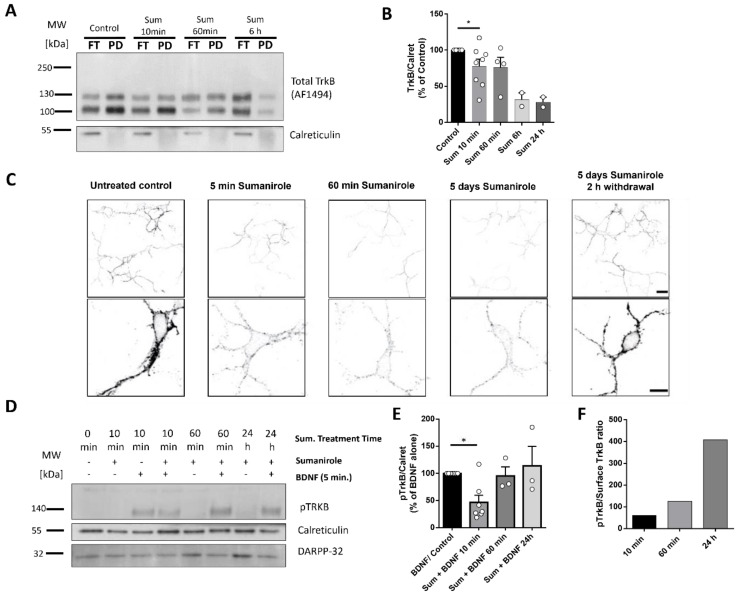
DRD2 activation with sumanirole decreases TrkB cell surface expression in iSPNs. (**A**) Western blot analysis of cell surface biotinylated flow cytometry-enriched DRD2-positive neurons at DIV 7-8 stimulated with sumanirole maleate (Sum). TrkB cell surface expression decreases gradually over time after stimulation of DRD2 for 10 min, 60 min, 6 h, and 24 h (see Appendix A). FT—flow through; PD—pull down. (**B**) Quantification of band intensity for TrkB cell surface expression, as shown in A. Data are shown as the mean ± SEM in % of untreated controls; Statistical analysis was performed using a one sample *t*-test using a mean of 100 (control). Number of replicates per condition are indicated in the figure as single data points. A confidence level of *p* ≤ 0.05 was set. *p*-values are indicated in the text and significant *p*-values are indicated in the figure. (**C**) Cell surface TrkB immunostaining of DRD2-eGFP-enriched neurons (DIV7) treated with sumanirole for 5 min, 60 min, and 5 days, and withdrawal for 2 h after 5 days of treatment compared against untreated controls maintained in the dopamine-depleted conditions. Scale bars: 20 µm (upper panel) and 10 µm (lower panel). (**D**) Western blot analysis of cell surface pTrkB induction of flow cytometry-enriched DRD2-positive neurons at DIV 7–8 stimulated with sumanirole maleate (Sum). TrkB phosphorylation is not observed under the control conditions or with sumanirole alone (lanes 1 and 2). Incubation with 10 ng/mL BDNF for 5 min induces TrkB phosphorylation (lane 3). Preincubation with 500 nM sumanirole maleate for 5 min followed by 10 ng/mL BDNF for another 5 min causes a significant reduction in pTrkB. Reduced pTrkB levels are not seen at later stages of preincubation with sumanirole at 60 min and 24 h. (**E**) Quantification of results from Western blots, as shown in D, from 3 independent experiments (n = 3). Statistical analysis was performed using a one-sample *t*-test. The control was set to 100%. Data are shown as the mean ± SEM in % of BDNF alone (n numbers are indicated in the graph for each condition as individual data points). A confidence level of *p* ≤ 0.05 was set. * *p* ≤ 0.05; empty circles represents individual data points All *p*-values are indicated in the text. (**F**) Ratio of pTrkB to cell surface-exposed TrkB in samples incubated with sumanirole for 10 min, 60 min, and 24 h (mean values of pTrkB/Surface TrkB: 10 min, 60.1; 60 min, 125.8; 24 h, 407.9.

**Figure 3 biology-12-01360-f003:**
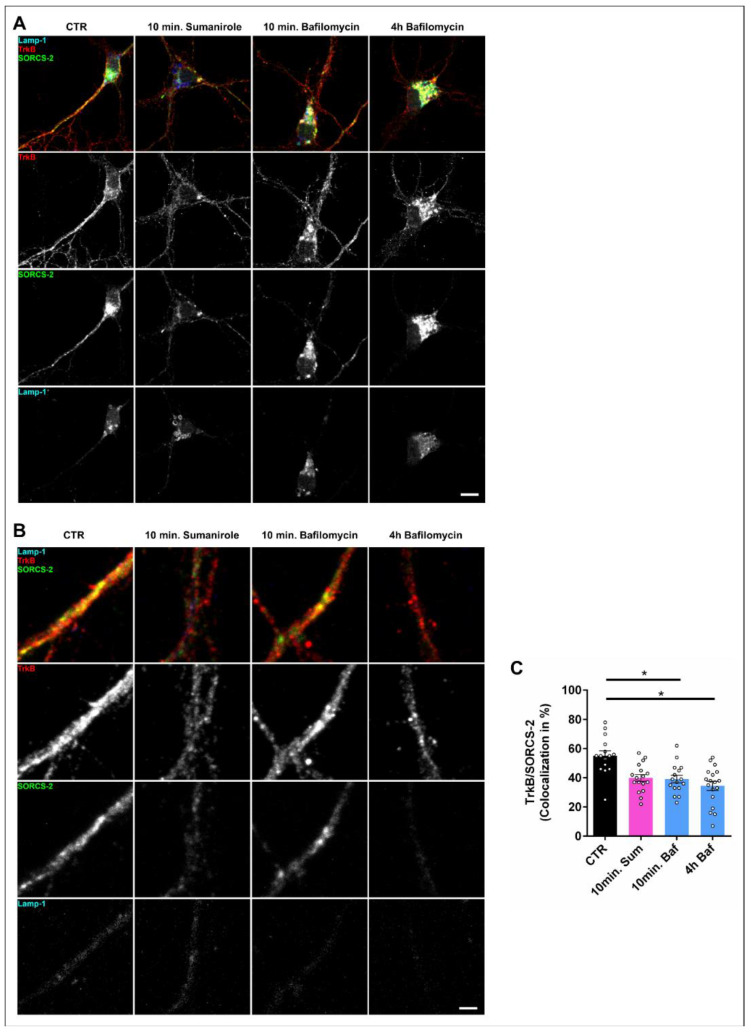
DRD2 activation with sumanirole removes TrkB from the peripheral dendrites in iSPNs. (**A**) TrkB and SORCS-2 immunoreactivity in purified DRD2-SPNs. Neurons were stimulated with sumanirole for 10 min or bafilomycin for 10 min or 4 h and compared with untreated controls. DRD2 activation with sumanirole reduces surface TrkB expression in iSPNs. Prolonged treatment with bafilomycin leads to an accumulation of SORCS-2- and Lamp-1-positive structures in the soma but not in the dendrites of iSPNs. Scale bar: 10 µm. (**B**) TrkB immunoreactivity in the peripheral dendrites of DRD2 neurons. Sumanirole treatment leads to a retraction of TrkB from the peripheral dendrites and less association with the cargo receptor SORCS-2. The bafilomycin treatment also decreased TrkB/SORCS-2 association. Scale bar: 2 µm. (**C**) Quantification of TrkB/SORCS-2 colocalization in peripheral dendrites stimulated with sumanirole (Sum) or bafilomycin (Baf). Data are shown as the mean ± SEM in % of TrkB/SORCS-2 colocalization. n = 3 for every condition. One-way ANOVA with Tukey’s multiple comparisons test. * *p* < 0.05.

**Figure 4 biology-12-01360-f004:**
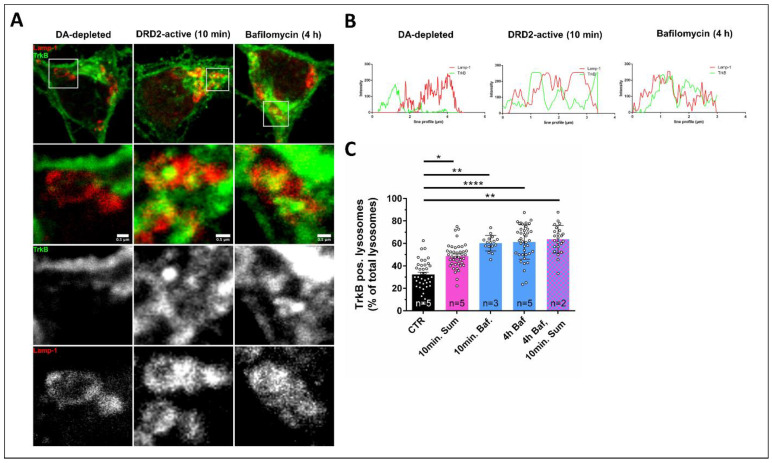
DRD2 activation with sumanirole leads to Lamp-1-associated intracellular TrkB clusters. (**A**) High magnification confocal microscopy shows that in iSPNs, TrkB is located at the plasma membrane. DRD2 activation induces TrkB’s retraction from the cell surface and the inhibition of its translocation to the cell surface. These images show the formation of intracellular TrkB clusters after the activation of DRD2, which are closely associated with Lamp-1-positive structures. (**B**) Exemplary line scans of TrkB and Lamp-1 intensities, as shown in A. (**C**) Quantification of TrkB/Lamp-1 structures, as shown in A. TrkB shows an increased association with lysosomes after DRD2 activation compared with the DA-depleted control. Data are shown as the mean ± SEM in % of all Lamp-1-positive structures. Number of biological replicates (cultures) per condition are indicated in the figure. One-way ANOVA with Tukey’s multiple comparisons test. * *p* < 0.05; ** *p* < 0.01; **** *p* < 0.001.

**Figure 5 biology-12-01360-f005:**
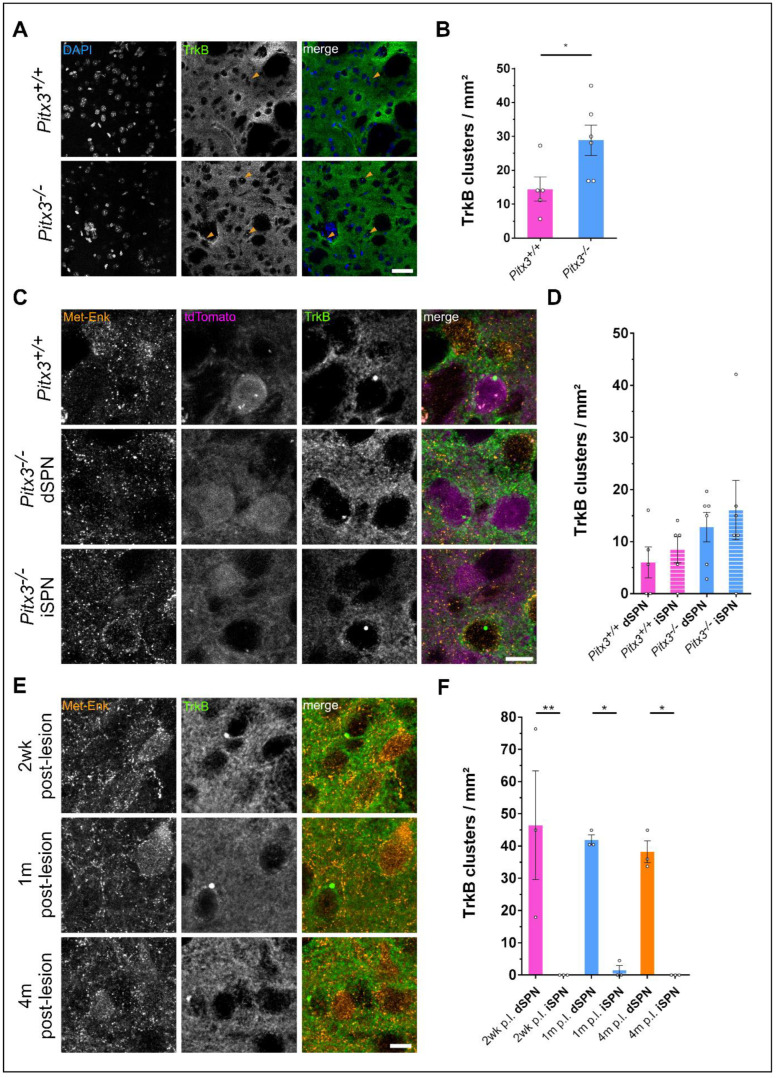
Dopamine depletion leads to TrkB cluster formation in striatal spiny projection neurons. (**A**) Upon dopamine depletion in the *Pitx3*^−/−^ mouse model, increased numbers of cytosolic TrkB clusters (marked by arrowheads) occur in the dorsolateral striatum. Scale bar: 40 µm. (**B**) Quantification of TrkB clusters, as shown in (**A**). TrkB clusters occur at a higher frequency in *Pitx3*^−/−^ than in wild-type control animals. Data are shown as the mean ± SEM. n = 5 for *Pitx3*^+/+^ and n = 6 for *Pitx3*^−/−^. Unpaired *t*-test (*p* = 0.0376). (**C**) TrkB clusters are found at perinuclear regions both in direct (tdTomato) and in indirect pathway (Met-Enk) spiny projection neurons. Scale bar: 10 µm. (**D**) Quantification of TrkB clusters, as shown in C. Data are shown as the mean ± SEM. n = 5 for *Pitx3*^+/+^ and n = 6 for *Pitx*^−/−^. One-way ANOVA with Tukey’s multiple comparisons test: *Pitx3*^+/+^ dSPN vs. *Pitx3*^−/−^ dSPN: *p* = 0.6254; *Pitx3*^+/+^ iSPN vs. *Pitx3*^−/−^ iSPN: *p* = 0.5326. (**E**) After 6-OHDA-induced lesion of the medial forebrain bundle in rats, the subsequent dopamine depletion leads to the formation of TrkB clusters in the dorsolateral striatum. These clusters occur exclusively in dSPNs, with no significant differences in their number at different time points after the lesion. dSPNs were identified as being negative for Met-Enkephalin, a marker for iSPNs. Scale bar: 10 µm. (**F**) Quantification of TrkB clusters shown in E. Data are shown as the mean ± SEM. n = 3 for every time point. One-way ANOVA with Tukey’s multiple comparisons test. * *p* < 0.05; ** *p* < 0.01.

**Figure 6 biology-12-01360-f006:**
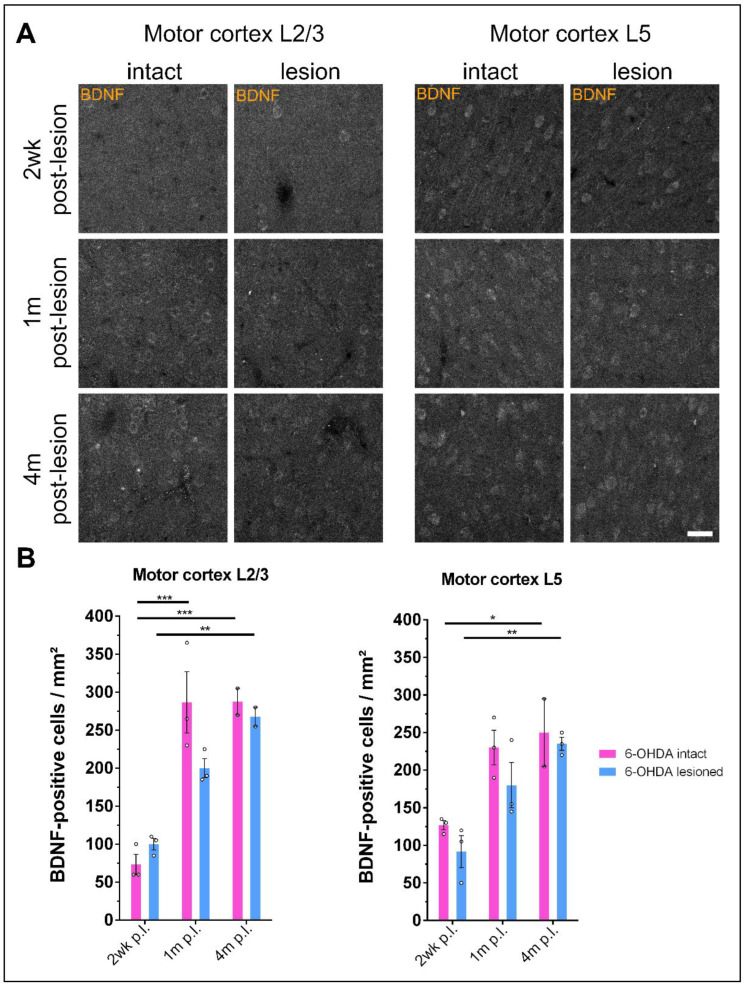
BDNF levels progressively increase in the motor cortex of 6-OHDA-treated rats. (**A**) After acute dopamine depletion using the 6-OHDA lesion model, rats show increasing BDNF expression in the motor cortex with prolonged time after the lesion both on the intact and lesioned side. Scale bar: 40 µm. (**B**) Quantification of BDNF expression, as shown in A. Data are shown as the mean ± SEM. n = 2–3 for every time point. One-way ANOVA with Tukey’s multiple comparisons test. * *p* < 0.05; ** *p* < 0.01; *** *p* < 0.001.

## Data Availability

Data is available upon request.

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
