# Peer review of "Dopaminergic Input Regulates the Sensitivity of Indirect Pathway Striatal Spiny Neurons to Brain-Derived Neurotrophic Factor"

_biology, 2023, doi:10.3390/biology12101360_

Round 1

Reviewer 1 Report

Ayon-Olivas and co-Authors are studying the TrkB distribution in response to DA and DA depletion in DRD2-expressing iSPNs. The authors demonstrated that activation of DRD2 in cultured striatal iSPNs and cholinergic interneurons causes a rapid retraction of TrkB from the cell surface, with a decrease of its sensitivity to BDNF. However, prolonged DRD2 stimulation restores baseline pTrkB levels, while surface TrkB is even more decreased. Using a mouse model of PD, the Pitx3-/-, they found perinuclear TrkB clusters both in dSPNs and in iSPNS.  In contrast, acute dopamine depletion in the 6-OHDA rat model leads to TrkB cluster formation in dSPNs but not in iSPNs.

In general, the study topic is interesting and, although the experimental design is good, the number of animals and samples are not enough. Thus the study should be improved. 

Comments:

-In the Results section, lines 488-89, and in the discussion section, line 587, references are missing. 

-In the Figure Legends, statistical results are partially described. The statistical analysis should be clarified. In all datasets, the authors should clearly indicate the F, degrees of freedom and significance for the various factors (time or session, treatment) and interactions. Please report all the non-statistically significant p values, they are still useful to know as a reader.

-The authors need to quantify the extent of the dopaminergic lesion in all groups to determine if lesions were similar in magnitude. They mentioned in the methods section the TH analysis, but data are missing. Moreover, the Pitx3 model has a denervation of 70%, as mentioned in the introduction lines 122-123, so is considered a partial model of PD compared with 6-OHDA fully lesion. Thus how the authors discuss the comparison between these two different models? Clarify this limit of the study.

-Globally, the number of samples (n, as reported in figure legends) in the different experiments is always rather small, in many cases even very small (n = 2-3), thus in these circumstances, the use of non-parametric tests is mandatory. This raise concerns as to whether these are sufficient statistical power to detect changes. This represents a huge limitation and if possible it would be necessary to increase the animals number or discuss this limitation in the text, to improve the quality and truthfulness of the work.

-In Figure 1A, the blot of Sum 24h is missing. Why? Moreover in B, in the quantification, statistical significances are not reported, as well as, the real number of the samples. Same consideration for Figure 6. Please, clarify. 

-In the Discussion section, authors mentioned dyskinesia, but they never explored this behavioral alterations. Furthermore, this section is too long and speculative, and many parts of the text don’t reflect the experiments done in this work. It is confused and with some concepts already mentioned in the introduction and results. This section must be revised and the limits of this study must be mentioned.

-Check the use of acronyms in the text and the abbreviations (e.g., SPN and not MSN).

Reviewer 2 Report

Dysfunction of the BDNF-TrkB pathway is known to be associated with neurodegenerative diseases, including Parkinson's disease. Here, the authors set out to investigate the reactivity of iSPNs for BDNF under both DRD2 activation and early dopamine depletion using the Pitx3 knock-out mouse model. The results indicate that DRD2 activation reduces TrKB cell surface levels in these neuronal populations. In contrast, prolonged DA depletion in the Pitx3 KO genetic mouse model causes the formation of TrKB clusters in both iSPNs and dSPNs. The data presented are exciting, also considering the consequences that these mechanisms may have in the case of prolonged treatment with levodopa.

I only have a few comments for the authors. 

1) As is well known, a population of projection spiny neurons in the striatum simultaneously express both D1 and D2 receptors. Did the authors consider investigating this neuronal population as well?

2) Why did the authors decide to investigate BDNF expression levels in motor cortex of 6-OHDA model instead of the Pitx3 KO genetic model, which is the one they focus the rest of the work on?

3)Although immunohistochemistry data also show the presence of DRD2-positive cholinergic interneurons in culture, it is unclear how activation of D2 receptors on these interneurons may contribute to TrKB internalization.

4) In discussions, the authors state that the increased phosphorylation of the TrKB receptor despite reduced membrane levels after prolonged DRD2 receptor activation may be explained by DRD2-positive astrocytes in sorted cells. Did the authors verify the presence of these DRD2-positive astrocytes and/or measure CNTF levels expression? 

5) In the introduction, the authors refer to "recent" studies on the role of the BDNF-TrKB pathway in LTP expression, but the suggested references include papers published between 1997 and 2010. 

Reviewer 3 Report

Ayon-Olivas et al. provide an interesting study to reveal that the activation of DRD2 in iSPNs decreases TrkB levels in plasma membrane. This study explains the opposite effects of DRD1 and DRD2 stimulation in synaptic plasticity. But there are several points should be addressed before publishing.

1. The (not) significant difference should be indicated in all graphs.

2. Figure 2A and 2B, the authors should provide at least three repeats for western blot.

3. Figure 2C should be explained as TrkB staining in figure legends.

4. In figure 3A, it is better to also provide the images of intact neurons instead of only dendrites, to show the the difference of TrkB, SORCS, Lamp1 under different conditions.

5. Figure 6 shows not difference in ipsilateral and contralateral to the 6-OHDA lesion. It can not be concluded that prolonged dopamine depletion enhances the BDNF expression. The results indicate the increase of BDNF levels is not related to 6-OHDA injection but correlated to aging. TH staining may better explain the DA depletion efficiency by 6-OHDA injection.

Round 2

Reviewer 1 Report

The authors have addressed all my concerns, and the suggestion of other revisors.

In my opinion the present version of the manuscript is significantly improved, and I find no obstacle to move it for publication.

Reviewer 3 Report

The authors satisfactorily address the questions.